



# Erosion risk assessment and identification of susceptibility lands

# using the ICONA model and RS and GIS techniques

Hossein Esmaeili Gholzom[1], Hassan Ahmadi[2], Abolfazl Moeini[1], Baharak Motamed Vaziri[1]

[1] Department of Forest, Range and Watershed Management, Faculty of Natural Resources and Environment, Science and Research Branch, Islamic Azad University, Tehran, Iran

[2] Department of Reclamation of Arid and Mountainous Regions, University of Tehran, Karaj, Iran

***Correspondence to:*** Abolfazl Moeini (abmoeini@yahoo.com)

**Abstract.** Soil erosion in Iran due to the destruction of natural resources has intensified in recent years and land use changes have played a significant role in this process. On the other hand, the lack of data in most watersheds to evaluate erosion and sedimentation for finding quick and timely solutions for watershed management has made the use of models inevitable. The purpose of this study was to use the ICONA model and RS and GIS techniques to assess the risk of erosion and to identify areas sensitive to water erosion in the kasilian watershed in northern Iran. The results of this study showed that with very high slope class percentage (20% - 35%) and sensitivity of shemshak formation to weathering which covers a large part of the watershed, soil erodibility class is high. But there is adequate land cover along with high percentage of natural forest cover, it has mitigated erosion. For this reason, the kasilian watershed is generally classified as low to moderate of erosion risk. Based on the erosion risk map, results show that the moderate class had the highest percentage of erosion risk (26.26%) at the watershed. On the other hand, the low erosion risk class comprises a significant portion (25.44%) of the catchment area. Also, 10.92% of the catchment area contains a very high erosion risk class, with most of it in rangeland and Rock outcrops second. However, the erodibility of the kasilian watershed is currently controlled by appropriate land cover, but the potential susceptibility to erosion is high. If land cover is redused due to inadequate land management, the risk of erosion is easily increased.

## 1 Introduction

Nowadays, with the growth and development of human activities, land use change, resource degradation and subsequent soil erosion are major problems in watersheds. This will, in the long run, obstacle the sustainable development of the environment, natural resources and agricultural lands. A study by Mohammadi et al. 2018 in Iran concluded that soil erosion in Iran has increased in recent years due to the destruction of natural landscapes. Understanding the extent of soil erosion risk in the absence of information in watersheds will enable critical areas of erosion to be identified. There is a lack of information in most of Iran's watersheds (Naderi et al., 2011). To achieve these goals, it is useful to use empirical models using RS and GIS techniques to estimate the sensitivity or potential of erosion risk. Numerous methods, including USLE, RUSLE, SIMWE, LISEM, QUERIM, PSIAC, MPSIAC, etc. have been used to predict and evaluate soil erosion and soil conservation planning. Qualitative assessment models based on the cognition that influence the factors affecting erosion can also play an important role in determining priorities affecting erosion and erosion susceptibility. One of these models is the ICONA model used in this study. Providing input data is a major problem that can be solved by remote sensing techniques and GIS analysis.

The use of RS and GIS techniques along with modeling processes such as soil erosion will accelerate the recognition, control and management of natural resources. GIS and RS make spatial data analysis faster and easier, and make it possible to combine extensive information across different fields and sources and simplify information management (Reis et al., 2017). In this situation, it is necessary to find quick and timely solutions. One of these solutions is the use of the ICONA model. This model was developed by the Spanish Society for the Conservation of Nature. Among many methods for predicting erosion using GIS and RS, simulation results of this model are widely accepted (Entezari, 2017).The ICONA model is one of the simplest and



most flexible qualitative methods for assessing and mapping soil erosion risk. This model is useful for describing and
comparing soil erosion in watersheds that do not have accurate and sufficient statistics. This model is an erosion risk assessment
method that utilizes qualitative decision rules and hierarchical organization of the four main inputs. This model is used in
Europe and Mediterranean countries (Okou et al., 2016). The erosion risk map prepared with the ICONA model can be a
reliable framework for erosion risk assessment (Zaz and Romshoo, 2012). This flexibility model can be used in decision-
making to solve erosion and destruction problems in the specific circumstances of each country or region (ICONA, 1997).
A case study carried out in the Bata watershed in Tunisia by Kefi et al. (2009) using the ICONA model and the use of RS and
GIS techniques showed that the Bata area, especially in areas with high slope and low vegetation cover, there is a very serious
problem of water erosion. Each watershed is also important in environmental, social and economic. In this regard, by managing
the erosion risk zoning and identifying the erodibility status of the watershed, management can be implemented to control and
reduce soil erosion (Olivares et al., 2011). However, sometimes the conditions of cover, rock facies and soil of some areas are
such that they limit the extent of erosion severity (Chatrsimab et al., 2017). A study by Sedighi (2011) in the Tangier-Red
watershed of Shiraz, Iran, using the ICONA model and the use of RS and GIS techniques. The results showed that the extent
of areas in the middle, high and high classes was increased during this time due to the change in land use. Karimi and Amin
(2012), in one study, zoned the erosion risk in Sivand Dam watershed in Fars province in Iran using the ICONA model and
RS technique. The results of this study showed that the watershed erosion rate has increased. They identified critical erosion
sites and proposed a management plan for it.
In this study, ICONA model was used to evaluate and determine the erosion risk status in kasilian watershed in northern Iran,
using RS and GIS techniques to determine the impact of factors affecting erosion. The ICONA model is a qualitative one, so
after completing the erosion risk mapping, we performed the model validation using the modified PSIAC method. In this study,
the soil erosion potential risk map with the ICONA model can be very important as a fast and practical method for soil
conservation decision makers and planners.

## 2 Data and methods

### 2.1 Study Area

The Kasilian Watershed is situated in the Mazandaran river watershed, one of the six major river watersheds in Iran.
Geographically, it lies within latitudes of 35° 58′ 45″ to 36° 07′ 45″ north, and longitudes 53° 01′ 30″ to 53° 17′ 30′ east (Fig.
1). The study area extends for about 6750 ha where the elevation ranges from 1100 to 2900 m.a.s.l. The area is characterized
by temperate climate according to De Martonne classification, while the Emberger climatic classification suggests a height
climate for the area (Hao and Aghakouchak, 2014; Hosseini pazhouh et al. 2018). According to a classification proposed by
the Natural Resources Conservation Service (NRCS), the hydrological soil group C well portrays the soil infiltration condition
of the study area in which a slow water infiltration and transmission rate prevails because the downward movement of the
water is impeded by moderately to very fine-textured soils. Also, forests, rangelands, farmlands, residential and rock outcrop
are the main land covers in the study area.



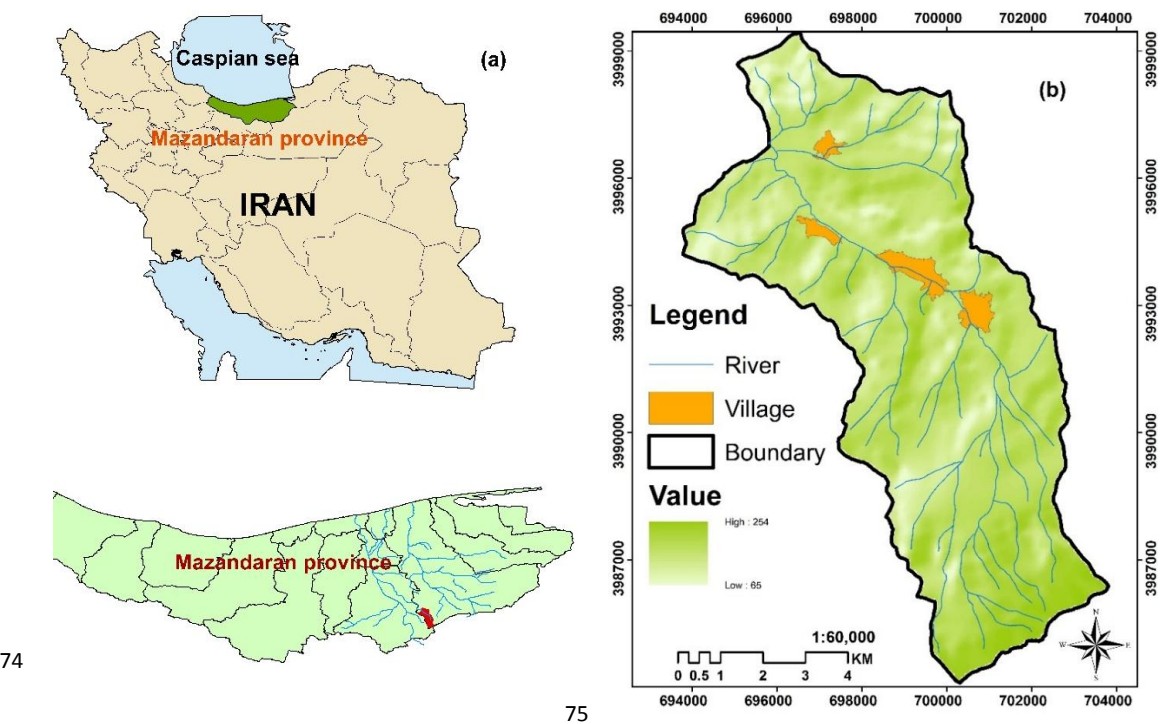

**Figure 1:** Geographical location of the kasilian watershed in Iran (a), Geographical coordinates system of kasilian watershed (b)
**2.2 Landsat data**
For this study, OLI satellite images with 30 m terrestrial resolution and spectral bands were used. Landsat satellite images of the
study area were produced in July 2017. These data are automatically referenced to the UTM coordinate system and the WGS 1984
elliptic system during ground harvesting by known coordinate points. However, the accuracy of the geometric correction of the
images was evaluated by overlaying the correlation data vector on the false color images of 4–3–3 and the topographic map of 1:
50,000 using Gaussian filtering. The average RMSE (Root Mean Square Error) error was estimated to be 0.48 pixel geometric
correction, which is acceptable. The two-step process proposed by Chander et al. (2009) was used to perform radiometric correction
of images. Atmospheric correction is performed using the FLAASH algorithm. This program corrects atmospheric effects during
SWIR and VNIR wavelengths. This program uses the standard equation for spectral radiation in the sensor, which is intended for
solar wavelength ranges (other than the thermal range) at the Lambert levels. Rewritten images were also transcribed using the
nearest neighbor interpolation method.
Training samples were prepared to map and then supervised classification was performed. Visual interpretation methods for images
and maps, Google Inheritance imagery, field visits and GPS pointers have been used for this purpose. More than 20 training and
control samples were selected for each user class. In total, 50% of the total number of samples were considered as control points. In
this study, Maximum likelihood method was used, which is the most suitable method for classification with supervision and its
classification results are produced as user maps.
**2.3 Modelling approach**
The ICONA model is a model developed and developed by the Spanish Institute of Natural Conservation (ICONA 1997; Bayramin
2003). It is a model for estimating the degree of erosion risk in watersheds that affect Its basis can be estimated at large scales of
erosion risk, which is applicable in European countries and many Mediterranean regions and is similar to many of the effective ways


to predict erosion using RS and GIS, the model was adopted in the above countries with similar climatic conditions (ICONA 1991).
The ICONA model consists of seven stages, as shown in Fig. 2 this is given.

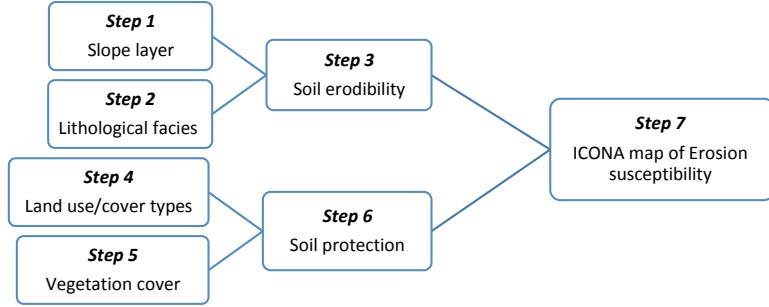


**Figure 2:** The methodological architecture of the ICONA model
Validation of the erosion risk map is an important aspect of model analysis. This can be done through quantitative evaluation
such as erosion measurement (Stroosnijder et al., 2003; Olivares et al., 2012; Kefi et al., 2009; Verieling et al., 2006). Due to the
empirical model of the ICONA model, to validate the results of this model with respect to its inputs, we evaluated the accuracy
of the ICONA model after preparing the erosion risk map using the modified PSIAC method. This method relies on the
calculation of various parameters including geology, soil, climate, surface erosion, slope, land cover, land erosion and gully
erosion. The erosion and sedimentation rate of the kasilian watershed after field laboratory operations were estimated according
to the modified PSIAC method. Validation of the erosion risk map is an important aspect of model analysis.
**2.4 Maps construction in ICONA model**
**2.4.1 Slope map**
In order to prepare the slope map of the studied watershed, the digital information of maps of 1:25000 Survey Organization[1] of
Iran was used. After preparing the digital elevation model (DEM) of the studied watershed, the slope map was obtained in
ArcGIS 10-3 environment. Then the watershed slope layer is produced in five classes: low and flat slope (%0-3%), medium
slope (3% -12%), high slope (%12-20%), very high slope (20% - 35%) And extremely high slope (more than> 35%).
**2.4.2 Lithofacies map**
Soil has been known to be the source of all subsequent developments in each area and it is very important to study the status of
the soil. The lithological units outcrop within the study area are classified according to physical and chemical resistance to
weathering (ICONA, 1997), which were classified into five groups. In this model we rely on different types of soil. Because
soils are often involved at the level of erosion processes, they constitute highly valuable and pivotal resources, and therefore our
classification should be based on the characteristics of the soil and the factor of soil erodibility (Okou et al., 2016). The amount
of soil erodibility factor was determined using USLE nomograph (Bayramin, 2003; Zaz and Romshoo, 2012). In this research
we used 1: 100000 maps of Iran Geological Organization[2].
**2.4.3 Erodibility map**
The soil erodibility layer was prepared by incorporating two layers of lithofacies and slope. The erodibility map indicates the

---

1 . The use of these maps as a basic map and scientific document is free.
2. The use of these maps as a basic map and scientific document is free.



potential (risk) of erosion in the watershed. In general, the integration of two slope and geological maps, according to the class
specificity of each map, constitutes 5× 5× 5 matrix, which in total is divided into 5 classes of Very Low (EN), Low (EB), Medium
(EM), High (EA) and very high (EX) are divisible (Table 1, Panel I).
**2.4.4 Land use/land cover map**
Satellite imagery and remote sensing techniques have been used to map the land. In this study, Landsat OLI satellite images with
30 m terrestrial resolution and spectral bands were used. For each user class, more than 20 training and control samples were
selected. In total, 50% of the total number of samples were considered as control points. In this study, Maximum Likelihood
method is used, which is the most suitable method for classification with supervision and its classification results are produced
as user maps (Tehrany et al., 2013 , 2014). Accuracy evaluation results are usually presented as an error matrix, in which case a
variety of parameters and values that indicate accuracy or some kind of error in the results are extracted from this matrix.
**2.4.5 Vegetation cover map**
Plants are illuminated in the range of 700 to 1,300 nm (near infrared) because they have a very high reflectance in the range,
reflecting the spectrum of green plants to extract vegetation mapping from the near-infrared split or band ratio process. Satellite
images are used for each pixel. Therefore, the Normal Vegetation Index (NDVI) provides information on the spatial and temporal
distribution of vegetation (photosynthesis) with photosynthetic activity and productivity (Tucker et al., 1985., Reed et al., 1994)
as well as the extent of land degradation in different ecosystems. It also shows (Holm et al., and Thiam, 2003). In this study, we
used this index for vegetation status and analyzed four classes of NDVI values for (1) low <25%, (2) moderate 25% - 50%, (3)
high 50% - 75% and (4) very high, more than> 75% (ICONA, 1997).
**2.4.6 Soil protection map**
At this stage, to obtain a soil conservation map, the overlay layer and vegetation layer overlap to form a 5× 5× 5 matrix. Soil
protection status class according to type of use and vegetation cover (MA) Very high protection, (A) high protection, (M)
moderate protection, (B) low protection and (MB) very low protection (Table 1, Panel II).
**2.4.7 Erosion susceptibility map**
In the last step of the ICONA model, a erosion risk map, soil conservation map and soil erodibility map were merged to overlay
the GIS. Consequently, according to Table 1, Panel III, according to the specificity of the classes of each map, they produce a
5× 5× 5 matrix which, in sum, has a map of erosion risk in the 5 erosion risk classes (1) very low, (2) low, ( 3) Moderate, (4)
high and (5) very high.












**Table 1:** Decision rule matrices for map overlapping

I

| Slope classes | Lithofacies | | | | |
|---|---|---|---|---|---|
| | a | b | c | d | e |
| Flat to gentle(%0-3%) | EN | EN | EN | EN | EB |
| Moderate (%3-12%) | EN | EN | EB | EM | EM |
| Steep(%12-20%) | EB | EB | EM | EA | EA |
| Very steep(%20-35%) | EM | EM | EA | EX | EX |
| Extremely steep(>35%) | EA | EA | EX | EX | EX |

II

| Land covers | Vegetation cover(%) | | | |
|---|---|---|---|---|
| | 0-25% | 25-50% | 50-75% | >75% |
| Rainfed farming | MB | MB | B | B |
| Irrigated farming | MB | MB | B | M |
| Forest | M | A | MA | MA |
| Orchard | B | M | A | MA |
| Rangeland | MB | B | M | A |
| Bare land | MB | M | A | MA |
| Rock outcrops | MA | MA | MA | MA |

III

| Soil protection | Soil erodibility | | | | |
|---|---|---|---|---|---|
| | EN | EB | EM | EA | EX |
| MA | 1 | 1 | 1 | 2 | 2 |
| A | 1 | 1 | 2 | 3 | 4 |
| M | 1 | 2 | 3 | 4 | 4 |
| B | 2 | 3 | 3 | 5 | 5 |
| MB | 2 | 3 | 4 | 5 | 5 |


**3 Results**
**3.1 Modelling steps**
The results regarding different calculation steps of the ICONA model are explained as follows.
**Step 1: Slope map**
According to Fig. 3a and Table 2, the bulk of the study area has a very steep slope (%20-35%) of 53.8%. The high slope class
(12% -20%) also comprises the second tier, equivalent to 20.2% of the area. Also, the extremely high slope ranks third (17.9%).
While the limited surface area of the watershed is low and flat.

**Table 2:** Areal percentages of the slope classes in the study area

| Slope classes | Slope(%) | Area(ha) | Area(%) |
|---|---|---|---|
| Flat to gentle | 0-3 | 26.98 | 0.4 |
| Moderate | 3-12 | 521.4 | 7.73 |
| Steep | 12-20 | 1363 | 20.2 |
| Very steep | 20-35 | 3632 | 53.8 |
| Extremely steep | >35 | 1207 | 17.9 |







**Step 2: Lithofacies map**
A map of the lithofacies of the kasilian watershed (Fig. 3b) shows that much of the watershed area, 72.6%, is composed of dark
gray and sandstone shale (Table 3). These soils are from Shemshak Formation and are of moderate to loose weathering. Also,
20.3% belong to Kashafrud Formation which are resistant to weathering.
The geological units of the area have a relatively wide range of permeability, with low permeability units having the most
surface expansion in the study area.

**Table 3:** Areal percentage of the lithofacies classes in the study area

| Classes | Materials and soil/rock resistance to weathering(lithofacies) | K factor | Area(ha) | Areal percentage in the study area |
|---|---|---|---|---|
| (a) Non-weathered | Conglomerate, heterogeneous sandstone, and shale with fossils and thin veins of coal (Kashafrud formation) | - | 1371 | 20.3 |
| (b) Fractured and/or medium weathered | Thick-bedded to massive light grey limestone (Lar formation) | $0.05<K<0.07$ | 377.9 | 5.6 |
| (c) Slightly to medium compacted | Dark grey shale and sandstone (Shemshak formation) | $0.1<K<0.2$ | 4898 | 72.6 |
| (d) Soft, low-resistant or strongly/deeply weathered | Grey to light green limestone with intercalations of calcite-shale (Dalichal formation) | $K\sim0.2$ | 57.32 | 0.85 |
| (e) Loose, non-cohesive sediment/soils and detritic material | Green-tuff with a heterogeneous assemblage of marine shale (Karage formation) | $K>0.6$ | 45.66 | 0.68 |


**Step 3: Erodibility map**
The soil erodibility map shows that 42.46% of the study area is highly erodible. Also, 30.58% of the watershed is in the class of
moderate erosivity and only 14.71% of the watershed is highly erodible. Fig. 3c and Table 4 show the erodibility status of the
study area.

**Table 4:** Areal percentage of the soil erodibility classes in the study area

| Classes | Label | Erodibility | Area(ha) | Areal(%) |
|---|---|---|---|---|
| 1 | EN | Very low | 186.36 | 2.761 |
| 2 | EB | Low | 641 | 9.496 |
| 3 | EM | Moderate | 2064.1 | 30.58 |
| 4 | EA | High | 2865.8 | 42.46 |
| 5 | EX | Very high | 992.78 | 14.71 |


192                                                      193



**Figure 3:** Maps of ICONA model, (a) slope map, (b) lithofacies map, (c) erodibility map





**Step 4: Land use/land cover map**
After classifying images and producing land use maps, the classification accuracy must be specified. For this purpose, kappa
coefficient, overall accuracy, user accuracy and producer accuracy were calculated (Table 5).

**Table 5:** An accuracy check of the classified land use/cover types

| Land use | Overall accuracy(%) | Kappa index | User's accuracy(%) | Producer's accuracy(%) |
|---|---|---|---|---|
| Forests | 82.26 | 0.75 | 91.15 | 90.6 |
| Farmlands | | | 73 | 65.38 |
| Rangelands | | | 62.58 | 85.71 |
| Residential | | | 59.17 | 76.09 |
| Rock outcrops | | | 63 | 66 |

The results of Fig. 4a and Table 6 show that the highest percentage of total land use was for forest use with 67.6% followed by
agricultural land. Minimum land use was also in residential areas with 2.22%.

**Table 6:** Areal percentage of the land use/cover classes in the study area

| Land use | Area(ha) | Areal(%) |
|---|---|---|
| Farmlands | 1512 | 22.4 |
| Forests | 4564 | 67.6 |
| Rangelands | 336 | 4.98 |
| Residential | 150 | 2.22 |
| Rock outcrops | 187 | 2.77 |


**Step 5: Vegetation cover map**
According to the results presented in Fig. 4b and Table 7, the highest percentage of vegetation based on high class NDVI index
(%50-75%) with 30.44% of area and the lowest vegetation percentage. According to this index, it belongs to the middle class
(%25-50%) with 19/63% of the total catchment area.

**Table 7:** Areal percentage of the vegetation cover classes in the study area

| Classes | Area(ha) | Areal(%) |
|---|---|---|
| Low(%0-%25) | 1851.6 | 27.43 |
| Moderate(%25-%50) | 1325 | 19.63 |
| High(%50-%75) | 2055 | 30.44 |
| Very high(>%75) | 1518.4 | 22.5 |


**Step 6: Soil protection map**
According to Fig. 4c and Table 8, 32.17% of the area has moderate protection. However, 30.91% of the area has very high
protection conditions. At the same time, only 12/11% of the area is in poor conservation conditions. Therefore, a significant
portion of the area is in good conservation conditions.

**Table 8**: Areal percentage of the soil protection classes in the study area

| Classes | Label | Soil protection | Area(ha) | Areal(%) |
|---|---|---|---|---|
| 1 | MA | Very high | 2086.5 | 30.91 |
| 2 | A | high | 967.67 | 14.34 |
| 3 | M | Moderate | 2171.3 | 32.17 |
| 4 | B | Low | 707.24 | 10.48 |
| 5 | MB | Very low | 817.28 | 12.11 |




225                                                                          226


228                          **Figure 4:** Protection map of ICONA model

**Step 7: Erosion risk map**
The middle class of erosion risk accounted for the largest percentage (26.26%) of the area. On the other hand, the low erosion
risk class comprises a significant portion (25.44%) of the catchment area. Therefore, a significant portion of the catchment area
is at moderate to low erosion risk. Only 10.92% of the area's surface constitutes a very high erosion risk class. The results of the
erosion risk map are shown in Fig. 5 and Table 9.




**Table 9:** Areal percentage of the soil erosion classes in the study area

| Classes | Erosion susceptibility | Area(ha) | Areal(%) |
|---------|------------------------|----------|----------|
| 1 | Very low | 1085.8 | 16.09 |
| 2 | Low | 1717.3 | 25.44 |
| 3 | Moderate | 1772.4 | 26.26 |
| 4 | High | 1437.5 | 21.3 |
| 5 | Very high | 736.97 | 10.92 |


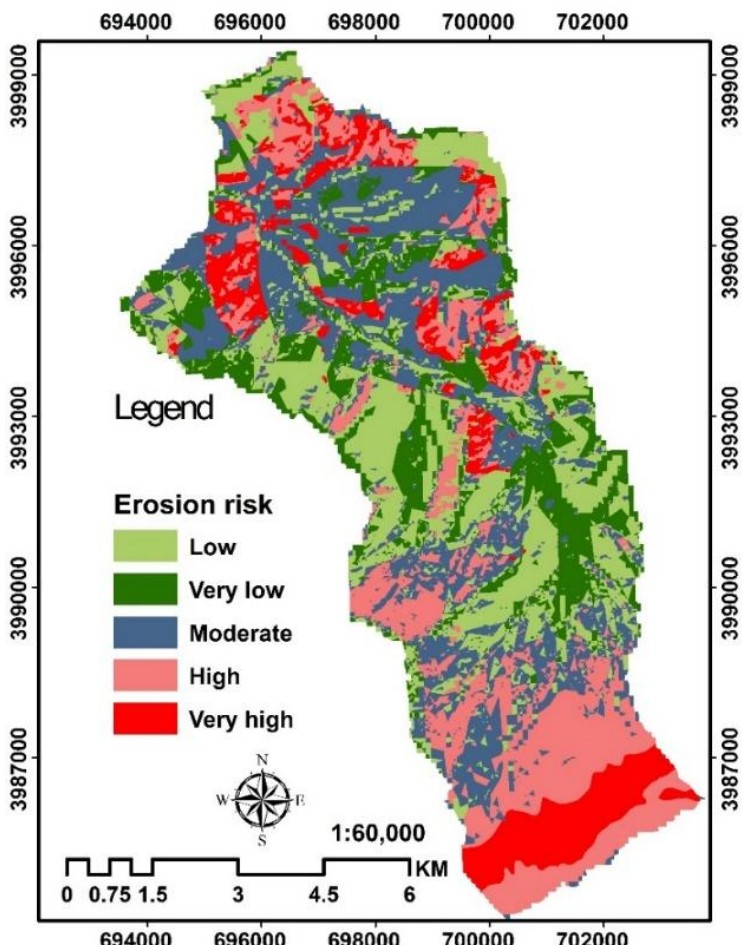

**Figure 5:** Erosion risk map of the ICONA model
**3.2 Results of land consolidation with risk of erosion**
After preparing the erosion risk map, we combined it with the land use map. The results of the integration are presented in Table
10. According to Table 10, the highest class of erosion risk in agricultural land is in the middle class (Fig. 6a).
According to Fig. 6b, the highest natural forest land was in the low class (20.1%) and the highest planted forest land was in the
low class (3.06%). On rangeland, almost %100 of this land use is in very high erosion class. This percentage actually accounts
for about 5% of the total study area (Fig. 6c). Rock outcrops comprise 99% of these lands and 2.74% of the total study area of
the watershed is in high risk of erosion (Fig. 6d). Residential land is also located within the arable land and generally accounts
for less than 1% of the total study area of erosion risk classes in this land use (Fig. 6e).

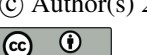



**Table 10:** Soil erosion–land use/cover matrix derived from the ICONA model

| Land use | Erosion susceptibility | Area(ha) | Percentage of total land use | Percentage of total study area |
|---|---|---|---|---|
| Farmlands | High | 322 | 21.3 | 4.77 |
| | Low | 125 | 8.26 | 1.85 |
| | Moderate | 694 | 45.9 | 10.3 |
| | Very high | 323 | 21.4 | 4.79 |
| | Very low | 48.1 | 3.18 | 0.71 |
| Natural forest | High | 891 | 19.5 | 13.2 |
| | Low | 1360 | 29.8 | 20.1 |
| | Moderate | 850 | 18.6 | 12.6 |
| | Very high | 36.9 | 0.81 | 0.55 |
| | Very low | 792 | 17.4 | 11.7 |
| | High | 45.9 | 1 | 0.68 |
| Planted forest | Low | 207 | 4.53 | 3.06 |
| | Moderate | 169 | 3.7 | 2.5 |
| | Very high | 31 | 0.68 | 0.46 |
| | Very low | 181 | 3.98 | 2.69 |
| Residential | High | 1.56 | 1.04 | 0.02 |
| | Low | 33.9 | 22.6 | 0.5 |
| | Moderate | 66 | 44 | 0.98 |
| | Very high | 6.98 | 4.65 | 0.1 |
| | Very low | 41.7 | 27.8 | 0.62 |
| Rangelands | High | 0.07 | 0.02 | 0 |
| | Low | 0 | 0 | 0 |
| | Moderate | 0.08 | 0.3 | 0 |
| | Very high | 336 | 100 | 4.98 |
| | Very low | 0 | 0 | 0 |
| | High | 185 | 99 | 2.74 |
| Rock outcrops | Low | 0 | 0 | 0 |
| | Moderate | 2.35 | 1.26 | 0.03 |
| | Very high | 0.01 | 0 | 0 |
| | Very low | 0 | 0 | 0 |














**Figure 6:** Overlay erosion risk with land use map, (a) agricultural land use, (b) forests land use, (c) rangeland land use, (d) rock outcrops landuse (e) residential land use

## 3.3 Validate the model

In the study area, there is a variety of erosion occurring, indicating the influence of different factors with different intensities and weaknesses along with the impact of human factors (Fig. 7a). These factors include the type of geological formations and their degrees of susceptibility to erosion, soil type, climate, surface currents, physiographic and topographic status, vegetation and river system type. Land use and how to observe or disrupt the proper rules and principles of operation, road construction and other construction operations also play a special role in the occurrence of various forms of erosion. As can be seen, the highest erosion intensity in the region is low to moderate and high erosion intensity states are not observed in the study area. On the other hand, surface erosion and rill erosion are the most important forms of erosion.

In this study, the validation of the ICONA model was compared with the current risk map and the current degradation map of the study area with the modified PSIAC model. Finally, the erosion intensity map (Fig. 7b) is prepared and compared with the





erosion risk map. In this study, the erosion and sedimentation rates were quantitatively and qualitatively determined using the
modified PSIAC method (Table 11). The study watershed with total scores of 53.7 and specific sediment yield of 332 tons /
km²/year with sum of scores of different factors can be said that the kasilian watershed is in the middle class in terms of erosion
class and in low grade in sedimentation.

**Table 11:** Quantitative status and quality of erosion by MPSIAC model

| The severity of the erosion | Sediment production t/km²/year | Quantitative evaluation of the effective factors on erosion |
|---|---|---|
| Intense | <2500 | 100> |
| Relatively intense | 1500-2500 | 75-100 |
| Moderate | 500-1500 | 50-75 |
| Low | 200-500 | 25-50 |
| Insignificant | 200< | 0-25 |


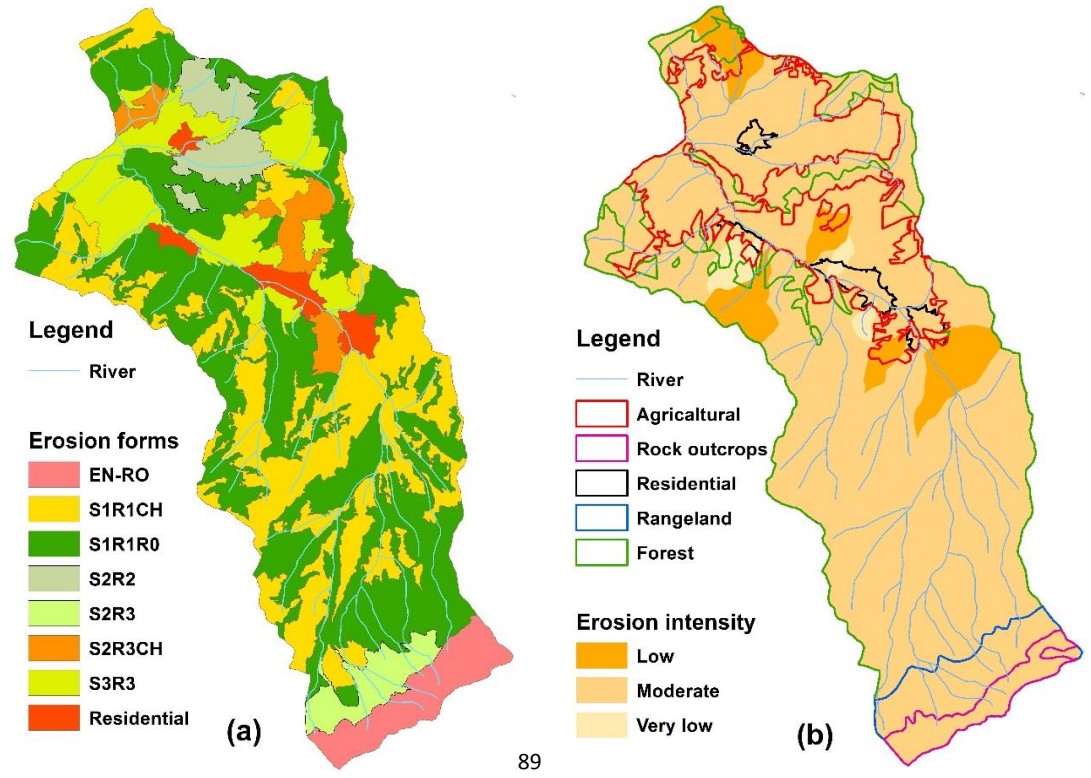

288                                                    89

290                 **Figure 7:** Maps of the study area by the modified PSIAC method, (a) erosion forms, (b) erosion intensity

R. rill erosion, S. surface erosion, EN. dissolution erosion, RO. rock mass loss, CH. channel erosion
**EN-RO.** Dissolution erosion and rock mass
**S₁R₁CH.** Surface, very low rill erosion along the river
**S₁R₁RO.** Surface, rill erosion with very low intensity and rock mass loss
**S₂R₂.** Surface and rill erosion with low intensity
**S₂R₃.** Low intensity surface and medium intensity rill erosion
**S₂R₃CH.** Low intensity surface, medium intensity riverbed and riverbank
**S₃R₃.** Surface and rill erosion of medium intensity


**4 Discussion**

The results show that the study area has a high slope percentage. Extreme slope class (%20-35%) with an area of 3632 ha (53.8% of total study area) has increased susceptibility to erosion (Fig. 3a and Table 2). The oldest geological units of the basin belong to lithofacies of Shemshak Formation and the most recent are alluvial sediments in the rivers of the region. The Shemshak Formation extends over 4800 ha in the basin. In terms of erodibility, most of the catchment area with the Shemshak Formation is located in relatively erodible units and comprises more than 72% of the area (Fig. 3b and Table 3).

The high slope and high surface area of the Shemshak Formation, which is sensitive to weathering, has increased the sensitivity and erodibility of the basin. The erodibility map of the basin indicates that a significant portion of the catchment area has high erosivity (42.46%) (Fig. 3c and Table 4). But according to Fig. 4c and Table 8, the presence of moderate (32.17%) and very high (30.91%) protective cover with high percentage of forest cover (67.60%) moderated the erosion. Forests are at low risk of erosion. This study shows the positive impact of natural vegetation on reducing erodibility and erosion risk by investigating land use in the kasilian watershed. In fact, areas with less vegetation suffer from more soil erosion (Lu et al., 2014; Uruk et al., 2012). According to Table 6, %22.4 of the soil surface cover is represented by various human uses with the threat of human erosion. The highest percentage of land use (45.9%) is in the middle erosion risk class (Table 10).

Also, according to Table 10, the results show that the highest and highest erosion risk classes, namely the areas most susceptible to erosion caused by agricultural operations in the study area, are 645 ha in total. Therefore, with operations in the field, it can be said that part of the agricultural activities in the steep slopes are unfavorable, which is very sensitive to erosion.

The findings showed that erosion is high in areas with high slope and low protection. This result is in agreement with the results of Kefi et al. (2009). This issue (impact of high slope and low protection) is also reported from the evidence of studies by Gatib and Larabi (2014) in Morocco and Volka et al. (2015) in Ethiopia working on the risk of erosion.

Although the climate conditions of the kasilian watershed are different from those, but in the kasilian watershed surveys, slopes of more than %35 constitute %17.9 of our study area. In these areas, the study area has a low risk of erosion on surfaces with high slope coverage. According to Okou et al. (2014) higher slope can also provide a natural protection against soil erosion. In higher elevation areas with more sensitive ecosystems such as grassland and rock outcrops, erosion-sensitive areas depend on soil status, slope, and type of land cover (Stanchi et al., 2013). By studying the kasilian watershed, the results show that the rangelands and outcrops in the upstream sections of the kasilian watershed are classified as high erosion (rock outcrop) and very high (rangeland) that have these special conditions (Table 10).

In the study area, there is a variety of erosion occurring, indicating the influence of different factors with different intensities and weaknesses along with the impact of human factors (Fig. 7a). The highest erosion intensity in the region is in the low to medium range and there is not much erosion. On the other hand, surface erosion and rill erosion are the most important forms of erosion in the area. Based on Table 11, with the assessment of erosion status at kasilian watershed, it can be concluded that the study area with a total score of 53.7 and specific sediment yield of 332 ton / km$^2$/year was qualitatively in moderate erosion class and in terms of sediment yield. The lower class is located (Fig. 7).

These results are in agreement with the results of the ICONA model. The highest percentage of erosion risk is in the wasilian watershed with the ICONA model in the middle class and in the second in the low class. This demonstrates the validity of using the ICONA model in the kasilian watershed.

**5 Conclusion**

The kasilian Watershed is located in the upstream areas of northern Iran with forest, agricultural, residential, pasture and rock outcrops. The majority of the study area (53.8%) has a very steep slope (%20 - 35%). Soils are susceptible to erosion in this basin. But by evaluating the ICONA model data, by evaluating other factors such as geological formations, vegetation cover and soil protection map, these factors have suitable conditions that can modulate the effective slope factor in erosion. This process



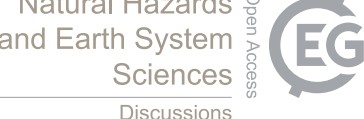

has reduced the area's sensitivity to erosion. Overall, the middle class of erosion risk accounted for the largest percentage
(26.26%) of the study area. The low erosion risk class (25.44%) also covers a significant portion of the catchment area. Only
%10.92 of the catchment area of the class is at high risk of erosion, with most of it in rangeland. Rangeland ranks first in terms
of risk of erosion. Rock outcrops are also classified as high risk of erosion. Field studies revealed that inappropriate farming
operations on steep slopes, excessive use of rangelands, and the existence of dissolution erosion in rock outcrops along with a
slope class greatly increased the sensitivity of these uses to erosion. It is suggested that more attention be paid to the study of
livestock grazing management and slope management. But most of the basin is not very eroded due to favorable conservation
conditions and suitable vegetation.
This study demonstrates that the erosion risk map prepared by the ICONA model using the RS and GIS techniques in the kasilian
watershed is sufficiently accurate. This model can be used as a reliable framework for erosion risk assessment and enables the
identification of potential erosion-prone areas. It can also be used as a watershed management approach for decision makers and
planners in watersheds as a fast and practical approach with reduced cost and time and good accuracy and capability utilizing
RS and GIS techniques.
**Acknowledgements.** We would like to thank the Forest, Range and Watershed Management Organization of Mazandaran
Province, Meteorological Organization of Mazandaran province for providing their invaluable data which has greatly
contribbuted to the completion of this work.
**Data availability.** Due to the nature of this research, the data used is mostly in the form of layers of maps that have been used
in the ICONA model. Erosion and sediment data have also been used to validate the model in the study area. This data can also
be viewed at http://hosseingholzam.blogfa.com.
**Author contributions.** HEG first suggested the subject of the research title. The research draft was then prepared and approved
by other authors (HA, AM and BMV). HEG prepared the initial data. During several stages of visiting the study area, the
necessary data were received from the field. AM analyzed satellite image data. BMV analyzed Numerical data. HA also did
model analysis and model validation. Finally, all authors contribute to the final analysis and final version of the article.
**Special issue statement.** This article is part of the special issue "Erosion risk assessment and identification of susceptibility
lands using the ICONA model and RS and GIS techniques". It is not associated with a conference.
**Competing interests.** The authors declare that they have no conflict of interest.

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
