# Peer review of "Erosion risk assessment and identification of susceptibility lands using the ICONA model and RS and GIS techniques"

_Natural Hazards and Earth System Sciences, 2020_

## Referee Comment (RC1) · Anonymous Referee #1 · 4 Aug 2020

General remarks This is a rather confusing manuscript for several reasons. The most important issue is the language. I advise the authors strongly to use a professional, native English translator for improving the language. Furthermore, the objective of this study is not clearly described. In the end, the erosion risk assessment using the ICONA model is compared with an existing result from the PSIAC model, without a discussion on the comparison. In 3.3, it is stated that high erosion intensity is not observed in the study area (line 277) (not clear where this is coming from), but no discussion on this outcome is made for discussing the results from the ICONA model.

I regret to state that this manuscript is unacceptable. If the authors present this in a

better way, indicating clearly the objectives, describe the validation of the results, put the results in perspective and discuss the consequences of the outcome in terms of possible focus areas for soil conservation, this may become an acceptable manuscript.

Specific remarks: 1. Line 31: present and describe the different models. You have listed here just a sequence of models without categorizing them in e.g. empirical versus process based, different timescales (event based, annual), different spatial scale. 2. Add more literature on erosion models and erosion risk assessment studies. 3. Line 38: why is it necessary to find 'quick and timely solutions'? 4. The introduction contains a lot of redundancies, try to funnel a bit more starting from the broad description of erosion models, risk assessment, the study site situation, and your objectives in this. 5. Section 2.1: I miss data of the climate for the study area. 6. Section 2.2: this sounds like a lot of work done for gathering the input data, but nowhere are results or a discussion of this procedure presented. Only the final map is presented. 7. Line 104: so you validate your model result with outcome of another model? That is no guarantee that results are reliable. 8. Table 1: Caption does not explain what is in the Table, what is MB, M, B, a, b, c etc.? 9. Table 3: what is K-factor? 10. Legend Fig 3: 'Very low' is lower than 'Low', so sequence should be adapted. Swap colours for 'Steep' and Very steep' (red is normally worse than light red) 11. Line 201-202: this procedure should be explained in 'methodology', and a discussion on the outcome should be presented. What do these figures imply? 12. Fig. 6: Colours and legend seem to be mixed here, use the same colours as for Fig 3. 13. Fig 7b: Legend is not in sequence of severity ('low-moderate-very low', should be 'very low-low-moderate'?) 14. Line 315: what is 'human erosion'? 15. Line 353: where is the conclusion that the erosion risk map is 'sufficiently accurate' based on? There is no ground prove (measured data) for this presented. If it is based on the comparison between models, then I would have expected a better discussion on that, and the limitations of this procedure explained.

---

## Author Comment (AC1) · 17 Aug 2020

dear referee. Thank you very much for your efforts. We hereby kindly appreciate your careful scrutiny on our submitted manuscript. It was tried to get your satisfaction by improving the manuscript. The following answers are offered to convince the respectful referees. We request you inform us if any further correction will be needed. I have marked the corrected items in red colour in the article.

Best regards Dr. moeini,

General remarks This is a rather confusing manuscript for several reasons. The most

important issue is the language. I advise the authors strongly to use a professional, native English translator for improving the language. Furthermore, the objective of this study is not clearly described. In the end, the erosion risk assessment using the ICONA model is compared with an existing result from the PSIAC model, without a discussion on the comparison. This article was revised by a native and professional English translator.

The purpose of this paper is to apply the ICONA model and using the GIS and RS techniques to assess the potential for erosion risk in a watershed in northern Iran and the southern part of the Alborz Mountains. To confirm the results of the ICONA model, while obtaining data and information from the basin by remote sensing, frequent visits to the basin and identifying the types of erosion and sensitive areas, the PSIAC model (as the base model) was used simultaneously. Finally, we matched the results of the ICONA model with the observations and data in the field and the results of the PSIAC model. The results of PSIAC model and ICONA qualitative model were highly consistent with field observations.

In 3.3, it is stated that high erosion intensity is not observed in the study area (line 277) (not clear where this is coming from), but no discussion on this outcome is made for discussing the results from the ICONA model. I regret to state that this manuscript is unacceptable. Yes, the Referee is right and this sentence (and high erosion intensity states are not observed in the study area) is added in line 277 and we removed it from the article.

If the authors present this in a C1 NHESSD Interactive comment Printer-friendly version Discussion paper better way, indicating clearly the objectives, describe the validation of the results, put the results in perspective and discuss the consequences of the outcome in terms of possible focus areas for soil conservation, this may become an acceptable manuscript. Specific remarks: 1. Line 31: present and describe the different models. You have listed here just a sequence of models without categorizing them in e.g. empirical versus process based, different timescales (event based, annual), different spatial scale. In the introduction, while correcting, we have provided information about the models (lines 30 to 37 and lines 40 to 45 and lines 68 to 71).

2. Add more literature on erosion models and erosion risk assessment studies. As stated in Section 1. From lines 30 to 37 , lines 40 to 45 and lines 68 to 71 more information was added about the models.

3. Line 38: why is it necessary to find 'quick and timely solutions'? According to the findings of Mohammadi, 2018 (line 26) and Naderi, 2011 ( line 29) of the soil erosion situation in Iran and the lack of sufficient data in most watersheds, it is necessary to find quick and timely solutions to assess the erosion situation. The ICONA model uses GIS and RS to perform erosion assessment in the watershed in less time and in a timely manner. This is continued in lines 50 and 51.

4. The introduction contains a lot of redundancies, try to funnel a bit more starting from the broad description of erosion models, risk assessment, the study site situation, and your objectives in this. According to the referee, red content has been added in introduction (lines 30 to 37 , lines 40 to 45 , lines 68 to 71and lines 72 to 82).

5. Section 2.1: I miss data of the climate for the study area. Some of the climate content of the study area was not mentioned, which is mentioned in section 1-2 and in line number 90 and 91).

6. Section 2.2: this sounds like a lot of work done for gathering the input data, but nowhere are results or a discussion of this procedure presented. Only the final map is presented. Yes, a lot of work has been done to collect input data. But a lot of the research data will be in the supplement and data availability section on line 374 and in the blog created.

7. Line 104: so you validate your model result with outcome of another model? That is no guarantee that results are reliable. Yes, you are right. As explained in the first part and section 15, the purpose of this sentence is to assessments the results of

the ICONA qualitative model compared to the PSIAC numerical model. However, the results of these models were matched with multiple field visits and erosion status assessments. That's why we delete the word credit.

8. Table 1: Caption does not explain what is in the Table, what is MB, M, B, a, b, c etc.? The description of these letters is given at the bottom of Table 1. Lines 181 to 187.

9. Table 3: what is K-factor? K-factor, shows the degree of soil erodibility in Table 3.

10. Legend Fig 3: 'Very low' is lower than 'Low', so sequence should be adapted. Swap colours for 'Steep' and Very steep' (red is normally worse than light red) We have changed the colors and legend map 3 .

11. Line 201-202: this procedure should be explained in 'methodology', and a discussion on the outcome should be presented. What do these figures imply? The contents of lines 201 and 202 are given in sections 2-2 and in lines 114 to 117. We have removed Table 5 and given the necessary numbers.

12. Fig. 6: Colours and legend seem to be mixed here, use the same colours as for Fig 3.

We have changed the colors and legend map 6 .

13. Fig 7b: Legend is not in sequence of severity ('low-moderate-very low', should be 'very low-low-moderate'?

We have changed the colors and legend map 7b .

14. Line 315: what is 'human erosion'? This term was modified in line 315, (Lines 319 and 320).

15. Line 353: where is the conclusion that the erosion risk map is 'sufficiently accurate' based on? There is no ground prove (measured data) for this presented. If it is based on the comparison between models, then I would have expected a better discussion on that, and the limitations of this procedure explained. We added these section in the

discussion section.

Numerous erosion forms have been identified through numerous studies and visits to the study area. PSIAC model with numerical data of 9 factors affecting erosion, the main purpose is to compare the results, not validation. The PSIAC model has been used (as a base model) in the field. In this study, the results of erosion risk assessment with the ICONA qualitative model are consistent with the map of erosion patterns and severity in the field.

---

## Short Comment (SC1) · 16 Sep 2020

I read the article "Erosion risk assessment and identification of susceptibility lands using the ICONA model and RS and GIS techniques" by Hossein Esmaeili Gholzom et al. (nhess-2020-85)" on the Natural Hazards and Earth System Sciences web site. For me, there are questions to improve the article that I sent. TOTAL COMMENT: Overall, the article addresses an interesting theme and uses an innovative and low-cost methodology that has produced satisfactory results in different regions of the world. The methodology is consistent and meets the requirements of the ICONA model. However, the article requires corrections and modifications before it can be published. In

this research, very good activities have been done with the data of the study area, information obtained from RS&GIS techniques to use the model. This article will be useful if the following comments are taken into consideration and corrected.

1- It is essential that the entire article be rewritten by a native English speaker. Abstract 2- In the abstract, the ICONA model method is mentioned. Give the ICONA model steps in two or three sentences. 3- Line 16 "Based on the erosion risk map, results show that the moderate class had the highest percentage of erosion risk (26.26%) at the watershed." Reword this sentence. Suggestion: Based on the erosion risk map, the results show that the moderate class had the highest percentage of erosion risk (26.26%) at the watershed level. Introduction 4- Line 52 "A study by Sedighi (2011) in the Tangier-Red watershed of Shiraz, Iran, using the ICONA model and the use of GIS & RS techniques. Reword this sentence Suggestion: A study by Sedighi (2011) in the Tangier-Red watershed in Shiraz, Iran, also used the ICONA model and GIS & RS techniques. 5- In the introduction, a better explanation should be given about the purpose of the research and the situation of the study area. Data and methods 6- In the study area section, provide information about rainfall and the annual temperature of the study area. 7- Line 95 correct "It is a model for estimating the degree of erosion risk in watersheds that affect Its basis can be estimated at large scales of erosion risk, which is applicable in European countries and many Mediterranean regions and is similar to many of the effective ways to predict erosion using RS and GIS, the model was adopted in the above countries with similar climatic conditions (ICONA 1991)." 8- Line 112, 113 Why you put the symbol % before the value? Correct in text, all figures and all tables. Results 9- The legend of maps numbers 3c, 4c, 5, 6, 7b should be modified to be very low, low, moderate, high and very high, respectively. 10- Table 2, the acronyms (EN, EB, MB, a, b, etc...) below the table 2. 11- Delete Table 5 and bring the effective coefficients such as kappa index and Overall accuracy percent in section 2.2 of the data and methods. 12- To evaluate the accuracy of the ICONA model, bring a combination of the erosion risk map with the erosion forms in the study area. 13- Section 3.3, if the use of the MPSIAC method (base model) is discussed, it is better
to compare the results of the ICONA model with the MPSIAC method that you did in the study area, and do not use the term validation with the MPSIAC model and delete this term. Discussion 14- Discussion should be properly organized according to the results. References 15- Line 428, correct the year of publication as other references.

---

## Author Comment (AC2) · 20 Sep 2020

Thank you very much for your efforts. We hereby kindly appreciate your careful scrutiny on our submitted manuscript. It was tried to get your satisfaction by improving the manuscript. The following answers are offered to convince the respectful referees. We request you inform us if any further correction will be needed. we have marked the corrected items in the article.

Best regards Dr. moeini,

Referee Comments Overall, the article addresses an interesting theme and uses an

innovative and low-cost methodology that has produced satisfactory results in different regions of the world. The methodology is consistent and meets the requirements of the ICONA model. However, the article requires corrections and modifications before it can be published. In this research, very good activities have been done with the data of the study area, information obtained from RSGIS techniques to use the model. This article will be useful if the following comments are taken into consideration and corrected.

1- It is essential that the entire article be rewritten by a native English speaker. - The article text was revised by an english language expert and Institute.

Abstract 2- In the abstract, the ICONA model method is mentioned. Give the ICONA model steps in two or three sentences. - The ICONA model process was presented in lines 6 and 7. 3- Line 16 "Based on the erosion risk map, results show that the moderate class had the highest percentage of erosion risk (26.26Reword this sentence. Suggestion: Based on the erosion risk map, the results show that the moderate class had the highest percentage of erosion risk (26.26- The Suggestion sentence was replaced in line 16.

Introduction 4- Line 52 "A study by Sedighi (2011) in the Tangier-Red watershed of Shiraz, Iran, using the ICONA model and the use of GIS  RS techniques.  Reword this sentence Suggestion: A study by Sedighi (2011) in the Tangier-Red watershed in Shiraz, Iran, also used the ICONA model and GIS  RS techniques. - The Suggestion sentence was replaced in line 52.  5- In the introduction, a better explanation should be given about the purpose of the research and the situation of the study area. - The research purpose in the study area was completed in lines 74 to 84.

Data and methods 6- In the study area section, provide information about rainfall and the annual temperature of the study area.  - Amount of rainfall and temperature are brought in lines 92 to 93. 7- Line 95 correct "It is a model for estimating the degree of erosion risk in watersheds that affect Its basis can be estimated at large scales of erosion risk, which is applicable in European countries and many Mediterranean

regions and is similar to many of the effective ways to predict erosion using RS and GIS, the model was adopted in the above countries with similar climatic conditions (ICONA 1991)." - This text was replaced in line 95. 8- Line 112, 113 Why you put the symbol - The

Results 9- The legend of maps numbers 3c, 4c, 5, 6, 7b should be modified to be very low, low, moderate, high and very high, respectively. - Map legend in maps 3c, 4c, 5, 6 and 7b was modified. 10- Table 2, the acronyms (EN, EB, MB, a, b, etc...) below the table 2. - I think it is Table 1, The acronyms (EN, EB, MB, a, b, etc...) are described below the table 1. 11- Delete Table 5 and bring the effective coefficients such as kappa index and Overall accuracy percent in section 2.2 of the data and methods. - Table 5 has deleted and brought the effective parameters in Section 2.2, lines 116 to 119. 12- To evaluate the accuracy of the ICONA model, bring a combination of the erosion risk map with the erosion forms in the study area. - Erosion risk situation, with field observations, remote sensing and models results are brought in Figure 7 (Maps 7a, 7b, 7c and 7d). 13- Section 3.3, if the use of the MPSIAC method (base model) is discussed, it is better to compare the results of the ICONA model with the MPSIAC method that you did in the study area, and do not use the term validation with the MPSIAC model and delete this term. - Line 268, Section 3.3 was modified. Also, in the Lines 274 to 276, lines 279 to 283 and lines 327 to 335 content was brought.

Discussion 14- Discussion should be properly organized according to the results. - The discussion section has been reviewed. This section was modified and properly organized.

References 15- Line 428, correct the year of publication as other references. - This reference is modified (line 440).

---

## Referee Comment (RC2) · Anonymous Referee #2 · 9 Oct 2020

I read the manuscript "Erosion risk assessment and identification of susceptibility lands using the ICONA model and RS and GIS techniques". The manuscript describes the application of remote sensing data, GIS and the erosion risk model ICONA in order to identify areas that might appear to be susceptible to soil erosion. After reading the manuscript, I see a lack of innovation and inconsistencies throughout the entire manuscript. In my opinion, it might be considerable for publication after the authors made some severe modifications and a major revision has been done. With kind regards

General comments - The manuscript is written in poor English and should be revised.

- The authors do not provide a research gap. The manuscript describes a case study, which is per se not a problem but I see a major lack in innovation. - The conclusions drawn partly do not reflect the outcomes of the approach. - The terms "erosion risk" and "erosion susceptibility" are confusingly and not consistently used throughout the entire manuscript.

Specific comments L12: RS and GIS were never abbreviated. L20: What is meant with "appropriate land cover"? L21: redused = reduced L21-22: This is a very broad and obvious statement. L24: This not just happens "nowadays". L30: The authors should be careful with the use of the terms "sensitivity" and "potential" in this case. L31: The authors are requested to provide references for the mention models. L35-40: The information provided in this paragraph can be condensed to a single sentence. L43: What are "the four main inputs"? L60: Reference for PSIAC method is required. L60-62: I do not see a scientific innovation or a research gap needed to be filled. L66: Coordinated do not have to be mentioned since they appear in the cross-referenced figure. Figure 1: The illustration of Mazandaran province poses another subplot and should be numbered as the others. The colour scheme from light green to green is rather not beneficial to illustrate elevation. L78-79: Reference for the data sets is requested. L90: Where were those samples collected? L94: "developed and developed". The description of the ICONA model was already mentioned in the introduction. These sentences are redundant. L104: How does the evaluation procedure works precisely? This is too general. L107: Well, validation might be important, but it is not performed in this paper and it cannot be done without ground truth data. L149: 5x5x5 matrix? Which quantitative values have these erosion risk classes? L231: The authors should not use the term "significant" if they did not perform a statistical analysis that provides information about statistical significance. L272-278: This is not a result. L304: Generating a classified slope map is a very limited finding.

———————————————

---

## Author Comment (AC3) · 29 Oct 2020

Subject: nhess-2020-85 – Author Comment Thank you very much for your efforts. We hereby kindly appreciate your careful scrutiny on our submitted manuscript. It was tried to get your satisfaction by improving the manuscript. The following answers are offered to convince the respectful referees. We request you inform us if any further correction will be needed. Modified items are marked in the new revised manuscript.

Best regards Dr. moeini,

Anonymous Referee 1

[Figure]

General remarks this is a rather confusing manuscript for several reasons. The most important issue is the language. I advise the authors strongly to use a professional, native English translator for improving the language. Furthermore, the objective of this study is not clearly described. In the end, the erosion risk assessment using the ICONA model is compared with an existing result from the PSIAC model, without a discussion on the comparison. In 3.3, it is stated that high erosion intensity is not observed in the study area (line 277) (not clear where this is coming from), but no discussion on this outcome is made for discussing the results from the ICONA model. I regret to state that this manuscript is unacceptable. If the authors present this in a C1 NHESSD Interactive comment Printer-friendly version Discussion paper better way, indicating clearly the objectives, describe the validation of the results, put the results in perspective and discuss the consequences of the outcome in terms of possible focus areas for soil conservation, this may become an acceptable manuscript.

Specific remarks: 1. Line 31: present and describe the different models. You have listed here just a sequence of models without categorizing them in e.g. empirical versus process based, different timescales (event based, annual), different spatial scale. - It is corrected. Introduction modified in the new revised manuscript and new contents were added, lines 32 – 47.

2. Add more literature on erosion models and erosion risk assessment studies. - In the introduction, more literature were added in the new revised manuscript, lines 48 – 65.

3. Line 38: why is it necessary to find 'quick and timely solutions'? - After modifying the introduction, This term was removed in the new manuscript. Our goal is to use a model that can describe the soil erosion state with minimum parameters, minimum time and low cost, fast and high accuracy with the use of RS/GIS techniques. This is possible with the ICONA model, Lines 64-65, 94 – 99.

4. The introduction contains a lot of redundancies, try to funnel a bit more starting from the broad description of erosion models, risk assessment, the study site situation, and

your objectives in this. - Introduction is modified and these cases were added in the new revised manuscript, lines 74 – 84.

5. Section 2.1: I miss data of the climate for the study area. - It is mentioned in section 1-2 in the new revised manuscript, lines 112 and 113.

6. Section 2.2: this sounds like a lot of work done for gathering the input data, but nowhere are results or a discussion of this procedure presented. Only the final map is presented. - In order to be concise and reduce the volume of the article, the product of the work done was used. Because the main purpose is to describe the ICONA model and its effectiveness in assessing the risk of erosion in the study area.

7. Line 104: so you validate your model result with outcome of another model? That is no guarantee that results are reliable. - Our goal is not to validate the ICONA model with the MPSIAC model. In most of the watersheds of Iran, the PSIAC model is widely used. For this reason, the purpose of providing the MPSIAC model is only as a base model to evaluate the results of the ICONA model with it. Then we can determine the future efficiency or inefficiency of the ICONA model in other watersheds. The necessary corrections to the new revised manuscript are given in lines 101-103 (introduction), 154-160 (data and methods), 296-309 (section 3.3) and figure 7, 358-365 (discussion) and 383-385 (conclusion). Section 3.3 has been completely revised and improved.

8. Table 1: Caption does not explain what is in the Table, what is MB, M, B, a, b, c etc.? - The description of these acronyms is given at the bottom of Table 1.

9. Table 3: what is K-factor? - K-factor, shows the degree of soil erodibility.

10. Legend Fig 3: 'Very low' is lower than 'Low', so sequence should be adapted. Swap colours for 'Steep' and Very steep' (red is normally worse than light red) - We have changed the colors and legend Fig 3. 11. Line 201-202: this procedure should be explained in 'methodology', and a discussion on the outcome should be presented. What do these figures imply? - Table 5 was deleted and The necessary information

was given in Section 2.2, lines 138-141.

12. Fig. 6: Colours and legend seem to be mixed here, use the same colours as for Fig 3.

- We have changed the colours and legend Fig 6. 13. Fig 7b: Legend is not in sequence of severity ('low-moderate-very low', should be 'very low-low-moderate'? We have changed the colours and legend Fig 7. 14. Line 315: what is 'human erosion'? - This sentence was removed by modifying and organizing the discussion section in the new revised manuscript. It refers to human activities that exacerbate erosion. Of course, this phrase is not translated correctly.

15. Line 353: where is the conclusion that the erosion risk map is 'sufficiently accurate' based on? There is no ground prove (measured data) for this presented. If it is based on the comparison between models, then I would have expected a better discussion on that, and the limitations of this procedure explained. - In order to explain this issue, The necessary corrections to the new revised manuscript are given in lines 101-103 (introduction), 154-160 (data and methods), 296-309 (section 3.3) and figure 7, 358-365 (discussion) and 383-385 (conclusion).

---

## Author Comment (AC4) · 29 Oct 2020

Subject: nhess-2020-85 – Author Comment Thank you very much for your efforts. We hereby kindly appreciate your careful scrutiny on our submitted manuscript. It was tried to get your satisfaction by improving the manuscript. The following answers are offered to convince the respectful referees. We request you inform us if any further correction will be needed. Modified items are marked in the new revised manuscript.

Best regards Dr. moeini,

SC1 Comments, Received and published: Majid Nozari, 16 Sep 2020

[Figure]

Overall, the article addresses an interesting theme and uses an innovative and low-cost methodology that has produced satisfactory results in different regions of the world. The methodology is consistent and meets the requirements of the ICONA model. However, the article requires corrections and modifications before it can be published. In this research, very good activities have been done with the data of the study area, information obtained from RSGIS techniques to use the model. This article will be useful if the following comments are taken into consideration and corrected. Specific remarks: 1- It is essential that the entire article be rewritten by a native English speaker. - The article text was revised by an English language expert and Institute.

Abstract 2- In the abstract, the ICONA model method is mentioned. Give the ICONA model steps in two or three sentences. - The ICONA model process was presented in the new revised manuscript, lines 13 – 15.

3- Line 16 "Based on the erosion risk map, results show that the moderate class had the highest percentage of erosion risk (26.26Reword this sentence. Suggestion: Based on the erosion risk map, the results show that the moderate class had the highest percentage of erosion risk (26.26- The Suggestion sentence was replaced in lines 18 and 19.

Introduction 4- Line 52 "A study by Sedighi (2011) in the Tangier-Red watershed of Shiraz, Iran, using the ICONA model and the use of GIS  RS techniques.  Reword this sentence Suggestion: A study by Sedighi (2011) in the Tangier-Red watershed in Shiraz, Iran, also used the ICONA model and GIS  RS techniques. - The Suggestion sentence was replaced in lines 86 – 87.  5- In the introduction, a better explanation should be given about the purpose of the research and the situation of the study area. - We modified the manuscript and improved its literature. The purpose of this investigating is to use a model that can describe the erosion risk assessment with minimum parameters, minimum time, low cost and high accuracy by use to RS/GIS techniques. This is possible with the ICONA model. Lines 64-65, 92 – 99.
Data and methods 6- In the study area section, provide information about rainfall and the annual temperature of the study area. - Amount of rainfall and temperature are brought in lines 112 -113.

7- Line 95 correct "It is a model for estimating the degree of erosion risk in watersheds that affect Its basis can be estimated at large scales of erosion risk, which is applicable in European countries and many Mediterranean regions and is similar to many of the effective ways to predict erosion using RS and GIS, the model was adopted in the above countries with similar climatic conditions (ICONA 1991)." - This text was replaced in lines 143 - 146.

8- Line 112, 113 Why you put the symbol - The

Results 9- The legend of maps numbers 3c, 4c, 5, 6, 7b should be modified to be very low, low, moderate, high and very high, respectively. - Map legend in maps 3c, 4c, 5, 6 and 7b was modified. Also, the entire manuscript was modified.

10- Table 2, the acronyms (EN, EB, MB, a, b, etc...) below the table 2. - I think it is Table 1, The acronyms (EN, EB, MB, a, b, etc...) are described below the table 1.

11- Delete Table 5 and bring the effective coefficients such as kappa index and Overall accuracy percent in section 2.2 of the data and methods. - Table 5 has deleted and brought the effective parameters in Section 2.2, lines 138 - 141.

12- To evaluate the accuracy of the ICONA model, bring a combination of the erosion risk map with the erosion forms in the study area. - Erosion risk situation, with field observations, remote sensing and models results are brought in Figure 7 (7a, 7b, 7c and 7d).

13- Section 3.3, if the use of the MPSIAC method (base model) is discussed, it is better to compare the results of the ICONA model with the MPSIAC method that you did in the study area, and do not use the term validation with the MPSIAC model and delete this term. - Section 3.3 has been completely revised and improved. In order to explain

this issue, The necessary corrections to the new revised manuscript are given in lines 101-103 (introduction), 154-160 (data and methods), 296-309 (section 3.3) and figure 7, 358-365 (discussion) and 383-385 (conclusion).

Discussion 14- Discussion should be properly organized according to the results. - The discussion section has been reviewed. This section was modified and properly organized.

References 15- Line 428, correct the year of publication as other references. - This reference is modified, line 507.

---

## Author Comment (AC5) · 29 Oct 2020

Subject: nhess-2020-85 – Author Comment Thank you very much for your efforts. We hereby kindly appreciate your careful scrutiny on our submitted manuscript. It was tried to get your satisfaction by improving the manuscript. The following answers are offered to convince the respectful referees. We request you inform us if any further correction will be needed. Modified items are marked in the new revised manuscript.

Best regards Dr. moeini,

Anonymous Referee 2

[Figure]

I read the manuscript "Erosion risk assessment and identification of susceptibility lands using the ICONA model and RS and GIS techniques". The manuscript describes the application of remote sensing data, GIS and the erosion risk model ICONA in order to identify areas that might appear to be susceptible to soil erosion. After reading the manuscript, I see a lack of innovation and inconsistencies throughout the entire manuscript. In my opinion, it might be considerable for publication after the authors made some severe modifications and a major revision has been done.

General comments:

- The manuscript is written in poor English and should be revised.

The manuscript was revised by an English language expert and Institute.

- The authors do not provide a research gap. The manuscript describes a case study, which is per se not a problem but I see a major lack in innovation.

We modified the manuscript and improved its literature. The purpose of this investigating is to use a model that can describe the erosion risk assessment with minimum parameters, minimum time, low cost and high accuracy by use to RS/GIS techniques. This is possible with the ICONA model. Lines 64-65, 94 – 99.

- The conclusions drawn partly do not reflect the outcomes of the approach.

We have modified on the discussion and conclusion sections in the new revised manuscript.

- The terms "erosion risk" and "erosion susceptibility" are confusingly and not consistently used throughout the entire manuscript.

These terms have been further refined in the new revised manuscript edited by an English language expert.

Specific comments:

L12: RS and GIS were never abbreviated.

It is corrected on line 12 and throughout the new manuscript.

L20: What is meant with "appropriate land cover"?

"Appropriate land use plans" is what we replaced in line 22 in the new revised manuscript.

L21: redused = reduced

By modified and deleting lines 21 and 22, the term was removed in the new revised manuscript.

L21-22: This is a very broad and obvious statement.

We deleted this sentence in the new manuscript.

L24: This not just happens "nowadays".

We removed this term in the new revised manuscript.

L30: The authors should be careful with the use of the terms "sensitivity" and "potential" in this case.

These terms have been further refined in a new manuscript edited by an English language expert.

L31: The authors are requested to provide references for the mention models.

We have given the references in the new revised manuscript, lines 37-38.

L35-40: The information provided in this paragraph can be condensed to a single sentence.

These few lines are summarized in the new manuscript (lines 69-70).

L43: What are "the four main inputs"?

[Figure]

"This model starts with 4 layers of information (slope, geology, vegetation and land use)". We have given this in lines 13 - 14 and line 71.

L60: Reference for PSIAC method is required.

A reference is given in line 103 in the new revised manuscript.

L60-62: I do not see a scientific innovation or a research gap needed to be filled.

In the last paragraph of the introduction, we provided more correction and explanation.

L66: Coordinated do not have to be mentioned since they appear in the cross-referenced figure.

Yes, if you think I should remove coordinated, let us do it.

Figure 1: The illustration of Mazandaran province poses another subplot and should be numbered as the others. The colour scheme from light green to green is rather not beneficial to illustrate elevation.

We have modified the items in Figure 1.

L78-79: Reference for the data sets is requested.

A reference is given in line 126 of the new revised manuscript.

L90: Where were those samples collected?

These samples were prepared in different land uses of the study area.

L94: "developed and developed". The description of the ICONA model was already mentioned in the introduction. These sentences are redundant.

This sentence was removed in the new revised manuscript.

L104: How does the evaluation procedure works precisely? This is too general.

To explain the subject, the necessary corrections in the new manuscript are given in

Section 2.3, Lines 154-160 and section 3.3, Lines 296-309 and figure 7.

L107: Well, validation might be important, but it is not performed in this paper and it cannot be done without ground truth data.

Our goal is not to validate the ICONA model with the MPSIAC model. In most of the watersheds of Iran, the PSIAC model is widely used. For this reason, the purpose of providing the MPSIAC model is only as a base model to evaluate the results of the ICONA model with it. Then we can determine the future efficiency or inefficiency of the ICONA model in other watersheds. The necessary corrections to the new revised manuscript are given in lines 101-103 (introduction), 154-160 (data and methods), 296-309 (section 3.3) and figure 7, 358-365 (discussion) and 383-385 (conclusion). Section 3.3 has been completely revised and improved.

L149: 5x5x5 matrix? Which quantitative values have these erosion risk classes?

This matrix is related to erodibility classes and soil protection classes to prepare an erosion risk map.

L231: The authors should not use the term "significant" if they did not perform a statistical analysis that provides information about statistical significance.

Instead of the word "significant", we replaced the word "great".

L272-278: This is not a result.

Section 3.3 was completely modified in the new revised manuscript and improved with Figure 7.

L304: Generating a classified slope map is a very limited finding.

This sentence was corrected in the new revised manuscript (lines 328 - 329). The very high slope class covers an area of 3632 hectares. According to Table 2, a large part of the area of kasilian watershed is in the high to extremely high slope class (12-20, 20-35 and > 35), which covers a total area of 6202 hectares. That is why this sentence

was raised.